# Enhancing the Anti-Dispersion Capability of the AO-OFDM System via a Well-Designed Optical Filter at the Transmitter

Kai Lv [1,*], Chao Yu [2,*], Hao Liu [1], Anxu Zhang [1], Lipeng Feng [1], Xia Sheng [1], Yuyang Liu [1] and Xishuo Wang [1]

1 Research Institute of China Telecom Co., Ltd., Beijing 102209, China; zhanganx@chinatelecom.cn (A.Z.); fenglp@chinatelecom.cn (L.F.); liuyy26@chinatelecom.cn (Y.L.)
2 School of Information and Electronics, Beijing Institute of Technology (BIT), Beijing 100081, China
* Correspondence: lvkai@chinatelecom.cn (K.L.); yuchao@bit.edu.cn (C.Y.)

**Abstract:** This paper proposes a novel method to improve the anti-dispersion ability of the all-optical orthogonal frequency division multiplexing (AO-OFDM) system. By replacing the Sinc-shaped filter with a Gauss-shaped filter for sub-carrier generation and inserting a cyclic prefix (CP), the impact of dispersion on the system can be significantly mitigated. Formula derivation and numerical analysis of the pulse-shaping function of the AO-OFDM system in the time domain for each cycle indicated that the pulse-shaping function generated by the Gauss-shaped filter was less affected by the dispersion effect than that of the Sinc-shaped filter. Meanwhile, less inter-carrier crosstalk between carriers was also observed. After carrying out system transmission simulations employing these two different filters, we found that the AO-OFDM system based on the Gauss-shaped filter could greatly improve the anti-dispersion ability compared with the system based on a Sinc-shaped filter. When the parameter settings in both schemes were identical, that is, the number of subcarriers was 32 and the power of a single subcarrier was $-13$ dBm, the bit error rate (BER) of the system based on the proposed Gauss-shaped filter after 60 km SMF transmission was only $1.596 \times 10^{-3}$, while the BER of the traditional Sinc-shaped filter based system scheme was as high as $8.545 \times 10^{-2}$.

**Keywords:** all-optical orthogonal frequency division multiplexing; Sinc-shaped filter; chromatic dispersion; Gauss-shaped filter

## 1. Introduction

Due to higher spectrum utilization and more flexible carrier allocation strategies, the AO-OFDM system can be regarded as one of the core technologies of the next generation of optical networks with more flexibility and elasticity [1–6]. At present, AO-OFDM systems typically use optical arrayed waveguide gratings (O—AWGs) and wavelength-selective switches (WSSs) based on liquid crystal on silicon (LCoS) as the core components for generating orthogonal sub-carriers [7–12]. The WSS is a key component of a reconfigurable optical adding/dropping multiplexer and a key all-optical routing element in a wavelength-reconfigurable network. It can provide almost meshless switching and carrier assignment functions. The AO-OFDM system based on WSS can demodulate the carrier while separating and redirecting the subcarrier to a certain corresponding output port from the overlapping supercontinuum spectrum when performing carrier allocation through wavelength-selective switching [10–12]. Although there are other optical devices that can be used to implement AO-OFDM, such as AWG, multi-fiber Bragg gratings, and circuits with trees of delay line interferometers, these solutions still rely on the use of WSSs for subcarrier separation and flexible allocation [13,14]. Therefore, we believe that implementing AO-OFDM systems based on WSS is the most promising solution, as it can fully utilize the flexibility brought by LCoS.

In the AO-OFDM system, the LCoS—based WSS can be regarded as a slew of Sinc-shaped filters with multiple output ports. Finisar explained, for the first time, how to

control the splitting and recombination of input light by controlling the LCoS and then adding additional delay to produce multiple Sinc-shaped filters [15]. Controlling the liquid crystal to achieve the above function is to control the diffraction of the liquid crystal by loading the phase diagram generated by the complex algorithm on the liquid crystal. The main factor limiting the promotion of this scheme is the excessively high cost of LCoS. The complex liquid crystal phase diagram also requires a large number of optimization algorithms and digital signal processing (DSP), thus further increasing the application burden. Therefore, other AO-OFDM schemes also have great application potential in scenarios where economic costs need to be underlined and the carriers are not required to be flexibly allocated. The AO-OFDM system based on WSSs in Ref. [9] used 252 orthogonal sub-carriers to complete 857.4 km optical fiber transmission with a transmission rate of 10.08 Tbit/s. In Ref. [10], the data rate of AO-OFDM exceeded 26 Tbit/s, and the spectrum efficiency reached 6 bit/s/Hz. At the same time, Ref. [11] also reported that the AO-OFDM system based on AWG was applied to passive optical networks far exceeding Tbit/s. It should be pointed out that AO-OFDM systems, whether applied to large-capacity long-distance transmission or large-capacity passive optical networks, rely heavily on dispersion compensation. As reported in Refs. [7,16], the AO-OFDM system had very poor anti-dispersion ability and was limited to less than 20 km on a single-mode fiber (SMF) for the reason that chromatic dispersion caused the sub-carrier's intensity peaks to shift from the null points of the other sub-carriers, and the relative time delay among the subcarriers broke their orthogonality, which led to crosstalk.

In order to compensate for the fiber dispersion on the system, numerous solutions have been proposed, such as adding CP, fractional Fourier transformation, dispersion compensation fiber, and other optical domain dispersion compensation schemes, where dispersion monitoring is introduced at the receiving end to achieve adaptive dispersion compensation. The main problem of the above method is that the longer the transmission distance, the lower the accuracy of dispersion measurement. The dispersion compensation by the digital signal processing technique can also be used. The received signal can be transmitted in the frequency domain or time domain through inverse fiber transmission for dispersion compensation, but this kind of dispersion compensation method requires a high-performance analog-to-digital converter for an AO-OFDM system, and a digital sampling rate that exceeds the symbol rate by several times, making it unrealistic to be widely used in future cost-effective optical networks.

When CP is inserted into the AO-OFDM system, we believe that the optical filter, as a core component for generating orthogonal sub-carriers, can also play a larger role in enhancing the anti-dispersion capability of the AO-OFDM system. And changing the Sinc-shaped filter into a Gauss-shaped filter close to the main lobe of the Sinc function can also improve the anti-dispersion capability of the AO-OFDM system. For the WSS based on LCoS, it is much easier to load the phase diagram of the control LCoS to generate a Gauss-shaped filter with a variable center wavelength and adjustable bandwidth than a Sinc-shaped filter with the same flexibility. In Section 2, we will specifically analyze the differences in the implementation methods of the two filters and the orthogonality of the sub-carriers generated by the two filters. The simulation results show that the AO-OFDM system generated by the Gauss-shaped filter had high anti-dispersion ability.

## 2. Principle

The system setup of the WSS—based AO-OFDM system discussed in this article is shown in Figure 1. A mode-locked laser (MLL) is used as the laser source of the system, which shares a clock with the arbitrary waveform generator (AWG) to ensure that the time domain period of the supercontinuum generated by the high nonlinear fiber (HNLF) after spectral expansion is the same as the symbol period generated by the AWG. The supercontinuum signal modulated by the Mach-Zehnder modulator (MZM) first entered the LCoS—based WSS for filtering and was then divided into multiple sub-carriers at the output. We also adjusted the filter transfer function of this filter to improve the system's

anti-dispersion ability. After transmission through an optical fiber, the signal carried by each sub-carrier was demodulated by a photodetector (PD) after filtering through a Sinc-shaped filter at the receiving end. Our analysis of CP and Gauss-shaped filters that can improve the anti-nonlinear ability of the system was focused on the transmitting end. The right side of Figure 1 shows the frequency domain waveforms of the two filters and the corresponding eye diagrams after the CP with a size of 0.33 inserted in the ideal state. The number of subcarriers is 8, the blue one is the waveform based on the Sinc-shaped filter, and the red one is based on the Gauss-shaped filter. When the CP was not inserted, the AO-OFDM system based on the Sinc-shaped filter was better than that based on the Gauss-shaped filter because of the orthogonal transmission performance between the subcarriers. However, an AO-OFDM system that used a Gauss-shaped filter after the CP was inserted could also have good transmission performance and achieve error-free transmission when ignoring interference from other noises.

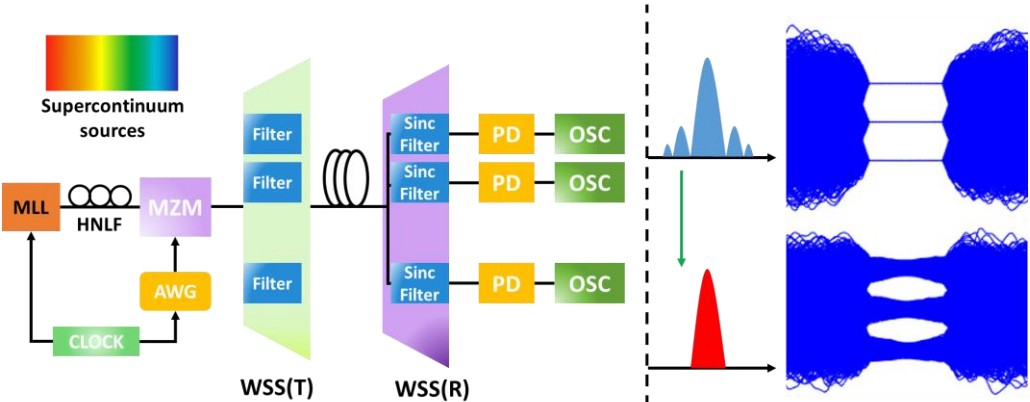

**Figure 1.** AO-OFDM system using Sinc-shaped filters as demultiplexers. On the right are the frequency-domain waveforms of the two filtering schemes and the eye diagrams after transmission under ideal conditions.

### 2.1. The Impact of CP on the AO-OFDM System

In the transmitter, the LCoS programmable WSS was used to perform Sinc-shaped filtering on a super-continuum optical source so as to generate multiple orthogonal carriers, which were further modulated to become the sub-carriers of the AO-OFDM system. These sub-carriers were multiplexed by WSS to form the complete AO-OFDM signal. Although the OFDM system with inserted CP required accurate chromatic dispersion compensation and monitoring at the receiving end, the CP could still mitigate the impact of dispersion on the AO-OFDM system, to a certain extent. We used the CP to cooperate with our proposed scheme to improve the system's resistance to dispersion.

In order to reduce the influence of lower inter-carrier interference and component defects on the system, the insertion of CP is essential. In a traditional electrical OFDM system, the CP is inserted by adding the end of each symbol to the beginning of the next symbol and approaches functions that can be easily implemented in the AO-OFDM system [10,17]. In Ref. [10], the CP was added by increasing the filter bandwidth of the filter that generated orthogonal subcarriers by supercontinuum filtering.

The signal of the AO-OFDM system with CP can be expressed as:

$$u_{OFDM}(t) = \sum_{i=-\infty}^{\infty} \sum_{k=0}^{N-1} C_{ki} u_k(t - iT_s) \tag{1}$$

$$u_k(t) = g(t) exp[j2\pi(\frac{k}{T_s} + f_c)t] \tag{2}$$

$$g_{cp}(t) = \begin{cases} 1 & (-\frac{T_b}{2} \leq t < \frac{T_b}{2}) \\ 0 & (other) \end{cases} \tag{3}$$

where $C_{ki}$ is the $i$th signal symbol on the kth subcarrier, $T_b$ is the symbol period, $f_c$ is the center frequency of the OFDM system, and $u_k(t)$ and $g_{cp}(t)$ are the waveform− and pulse-shaping functions of the *kth* subcarrier, respectively. The pulse-shaping function without CP is $g(t)$:

$$g(t) = \begin{cases} 1 & (-\frac{T_{OFDM}}{2} \leq t < \frac{T_{OFDM}}{2}) \\ 0 & (other) \end{cases} \tag{4}$$

where $T_s$ is the symbol period; the frequency domain filter functions corresponding to the two pulse-shaping functions are:

$$G_k(f) = sinc[(f - f_{sk})f_s] \tag{5}$$

$$G_{kcp}(f) = sinc[(f - f_{bk})f_b] \tag{6}$$

The added CP can be calculated as $cp = (f_b - f_s)/f_b$, where $f_b = 1/T_b$ and $f_S = 1/T_S$ are the bandwidth of the Sinc function and symbol rate, respectively. $f_{bk}$ is the center frequency of the Sinc-shaped filter corresponding to the kth orthogonal subcarrier with the filter bandwidth of $f_b$ after inserting CP. $f_{sk}$ is the corresponding center wavelength without CP being inserted. In the receiver, Sinc-shaped filtering with the same bandwidth as the WSS can be performed on the signal. When the bandwidth $f_b$ of the Sinc-shaped filter is greater than $f_s$, the CP will be inserted into each symbol period, as shown in Figure 2:

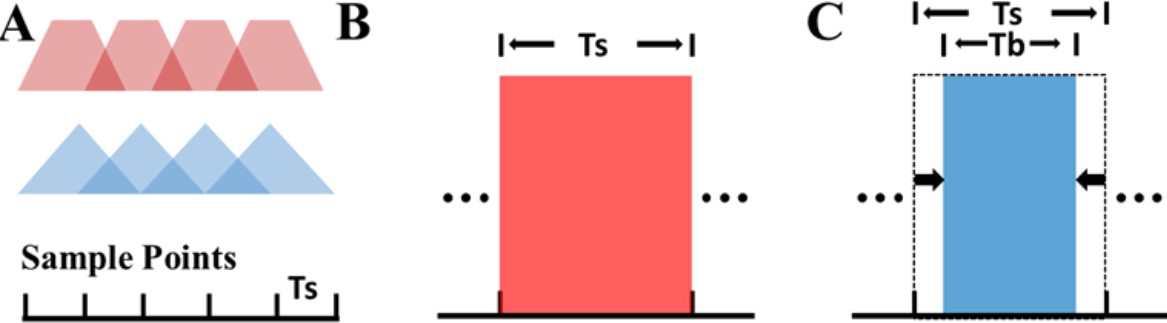

**Figure 2.** (**A**) The impact of CP on a single orthogonal sub-carrier; (**B**) time-domain pulse envelope of a single symbol without CP being inserted; and (**C**) single time-domain pulse envelope inserted into the CP.

The AO-OFDM signal with CP inserted at the receiving end can be expressed as follows through the corresponding Sinc-shaped filter at the receiving end:

$$u_{out}(t) = u_{OFDM}(t) * h_m(t) = \sum_{i=-\infty}^{\infty} \sum_{k=0}^{N-1} C_{ki} r_k(t - iT_s) \tag{7}$$

$$r_k(t) = \sum_{m=0}^{N-1} s_{mk}(t) \exp(j2\pi(\frac{k}{T_s} + f_c)t) \tag{8}$$

$$H_m(f) = sinc[(f - f_{bm})f_s] \tag{9}$$

$$s_{kk}(t)_{(m=k)} = \begin{cases} \frac{1}{T_s}\left(t - \frac{-T_s - T_b}{2}\right) & \left(\frac{-T_s - T_b}{2} \le t < \frac{-T_s + T_b}{2}\right) \\ \frac{T_b}{T_s} & \left(\frac{-T_s + T_b}{2} \le t < \frac{T_s - T_b}{2}\right) \\ \frac{1}{T_s}\left(-t + \frac{T_s + T_b}{2}\right) & \left(\frac{T_s - T_b}{2} \le t < \frac{T_s + T_b}{2}\right) \\ 0 & (other) \end{cases} \tag{10}$$

$$s_{mk}(t)_{(m \ne k)} = \begin{cases} \dfrac{\exp[j2\pi(\frac{k-m}{T_b})(t + \frac{T_s}{2})] - \exp[-j2\pi(\frac{k-m}{T_b})(\frac{T_b}{2})]}{j2\pi(\frac{k-m}{T_b})T_s} & \left(\frac{-T_s - T_b}{2} \le t < \frac{-T_s + T_b}{2}\right) \\ 0 & \left(\frac{-T_s + T_b}{2} \le t < \frac{T_s - T_b}{2}\right) \\ \dfrac{\exp[j2\pi(\frac{k-m}{T_b})\frac{T_b}{2}] - \exp[j2\pi(\frac{k-m}{T_b})(t - \frac{T_s}{2})]}{j2\pi(\frac{k-m}{T_b})T_s} & \left(\frac{T_s - T_b}{2} \le t < \frac{T_s + T_b}{2}\right) \\ 0 & (other) \end{cases} \tag{11}$$

where $H_m(f)$ represents the frequency domain function of the Sinc-shaped filter bank, $f_{bm}$ represents the center frequency of the $m$th Sinc-shaped filter at the receiving end, and $h_m(t)$ represents the corresponding time domain function. It is worth noting that the bandwidth of the filter at the receiving end is the same as the symbol rate. $r_k(t)$ and $s_{mk}(t)$ are the waveform and pulse-shaping function of the $k$th subcarrier. When $m$ is equal to $k$, that is, when the target subcarrier passes through the corresponding Sinc-shaped filter with the same center frequency, the pulse-shaping function is $s_{mk}(t)_{(m=k)}$. The adjacent orthogonal sub-carriers will also pass through the filter, and the pulse-shaping function is $s_{mk}(t)_{(m \ne k)}$ after filtering. Combining Equations (10) and (11), we can clearly see that, when the AO-OFDM system inserts CP, there is a flat range for each symbol period after filtering at the receiving end, and the flat range lasts for $T_s - T_b = 1/f_s - 1/f_b$. This approach will not destroy the orthogonality due to CP insertion and is not affected by other sub-carriers. The larger the filtering bandwidth of the Sinc-shaped filter at the transmitting end, the more CP is inserted. Although the larger the CP, the lower the spectrum utilization rate of the AO-OFDM system and the larger the flat interval in each symbol period, this interval can allow more sampling time errors.

## 2.2. Analysis of Anti-Dispersion Ability

We propose the use of a Gauss-shaped filter instead of a Sinc-shaped filter at the transmitting end can improve the system's anti-dispersion capability. As the orthogonal sub-carriers of the WSS—based AO-OFDM system are supercontinuum-filtered by the WSS, the frequency-domain function of the all-optical filter directly controls the frequency-domain waveform of the sub-carrier.

The AO-OFDM signal with CP generated by using a Gauss-shaped filter can be expressed as:

$$u'_{OFDM}(t) = \sum_{i=-\infty}^{\infty} \sum_{k=0}^{N-1} C_{ki} u'_k(t - iT_s) \tag{12}$$

$$u'_k(t) = g'_{cp}(t) e^{j2\pi(\frac{k}{T_s} + f_c)t} \tag{13}$$

$$g'_{cp}(t) = \sqrt{\frac{\pi}{2}} \cdot exp(-\frac{\pi^2 t^2}{2T_s^2}) \tag{14}$$

where $u'_k(t)$ and $g'_{cp}(t)$ are the waveform and pulse-shaping function of the $k$th subcarrier. The frequency domain waveform of the Gauss-shaped filter is:

$$G'_{kcp}(f) = exp\left[-\frac{2(f - f_{bk})^2}{f_s^2}\right] \tag{15}$$

where $f_{bk}$ is the center frequency of the Gauss-shaped filter corresponding to the $k$th subcarrier with the filter bandwidth of $f_s$ after inserting CP. Different from using a Sinc-shaped filter to insert the CP, the Gauss-shaped filter increased the sub-carrier spacing without increasing the filter bandwidth. As shown in Figure 3, the Gauss-shaped filter was highly similar to the Sinc-shaped filter with a bandwidth of $f_s$ when the intensity was greater than $-6$ dB.

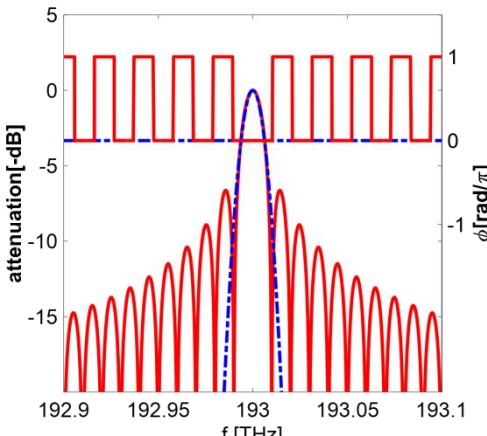

**Figure 3.** Comparison of the Sinc-shaped filter (red lines) and Gauss-shaped filter (blue lines) in the frequency domain.

The left axis in Figure 3 shows the amplitude response of the two filters' waveforms with frequency, and the right axis shows the phase response curve of the two filters with frequency. The blue dotted line represents the Gauss-shaped filter and the red solid line represents the Sinc-shaped filter. We first analyzed the orthogonality of the AO-OFDM system based on a Gauss-shaped filter under ideal conditions without being affected by other noises. Obviously, the carrier generated by the Gauss-shaped filter did not have strict orthogonality. There was crosstalk between carriers and crosstalk between symbols at the receiving end through the Sinc-shaped filter.

The AO-OFDM signal after passing through the Sinc-shaped filter at the receiving end can be expressed as:

$$u'_{out}(t) = u'_{OFDM}(t) * h_m(t) = \sum_{i=-\infty}^{\infty} \sum_{k=0}^{N-1} C_{ki} r'_k(t - iT_s) \tag{16}$$

$$r_k'(t) = \sum_{m=0}^{N-1} s'_{mk}(t) \exp\left[ j2\pi(\frac{k}{T_s} + f_c)t \right] \tag{17}$$

$$H_m(f) = sinc[(f - f_{bm})f_s] \tag{18}$$

$$s'_{kk}(t)_{(m=k)} = \begin{cases} \frac{1}{2} \cdot erf\left[\frac{\pi}{\sqrt{2}T_s}(t + \frac{T_S}{2})\right] - erf\left[\frac{\pi}{\sqrt{2}T_s}(\frac{T_S}{2} - t)\right] & (t \le -\frac{T_S}{2}) \\ \frac{1}{2} \cdot erf\left[\frac{\pi}{\sqrt{2}T_s}(t + \frac{T_S}{2})\right] + erf\left[\frac{\pi}{\sqrt{2}T_s}(\frac{T_S}{2} - t)\right] & (-\frac{T_S}{2} < t \le \frac{T_S}{2}) \\ \frac{1}{2} \cdot erf\left[\frac{\pi}{\sqrt{2}T_s}(t + \frac{T_S}{2})\right] - erf\left[\frac{\pi}{\sqrt{2}T_s}(\frac{T_S}{2} - t)\right] & (t > \frac{T_S}{2}) \end{cases} \tag{19}$$

$$s'_{mk}(t)_{(m \ne k)} = \frac{j}{2} \cdot \exp\left\{ \frac{-\left[2(m-k)^2 f_b^2\right]}{f_s^2} \right\} \left\{ erfi\left[ \frac{4(m-k)^2 f_b^2 + j\pi f_s(2f_s t - 1)}{2\sqrt{2}f_s} \right] - erfi\left[ \frac{4(m-k)^2 f_b^2 + j\pi f_s(2f_s t + 1)}{2\sqrt{2}f_s} \right] \right\} \tag{20}$$

The Sinc-shaped filter used at the receiving end of the two AO-OFDM systems is the same. If the same-sized CP is inserted, the filter function in the frequency domain is $H_m(f)$, and $r'_k(t)$ and $s'_{mk}(t)$ are the waveform and pulse-shaping function of the $k$th subcarrier,

respectively. When *m* is equal to *k*, that is, when the target subcarrier passes through the corresponding Sinc-shaped filter with the same central frequency, the pulse-shaping function is $s_{kk}(t)_{(m=k)}$. The other sub-carriers will also pass through the filter, and the pulse-shaping function is $s_{mk}(t)_{(m \neq k)}$ after filtering. $erf(\cdot)$ is the error function and $erfi(\cdot)$ is the imaginary error function. The sub-carriers that are different from the center frequency of the target filter will inevitably cause inter-carrier crosstalk to the target sub-carrier. Figure 4 shows the pulse-shaping functions $s_{kk}(t)_{(m=k)}$, $s_{mk}(t)_{(m \neq k)}$, $s'_{kk}(t)_{(m=k)}$, and $s'_{mk}(t)_{(m \neq k)}$ of the two systems with 0 CP inserted and the same subcarrier symbol rate at the receiving end after passing the same Sinc-shaped filter. These four pulse-shaping functions can clearly reflect the crosstalk between carriers after passing the filter. For convenience of comparison, we normalized the four pulse-shaping functions. We discuss the situation of transmitting "1" in an ideal state and receiving "1" after scaling at the receiving end. In the following analysis of the influence of chromatic dispersion on the AO-OFDM system, we also performed the same analysis on these four pulse-shaping functions.

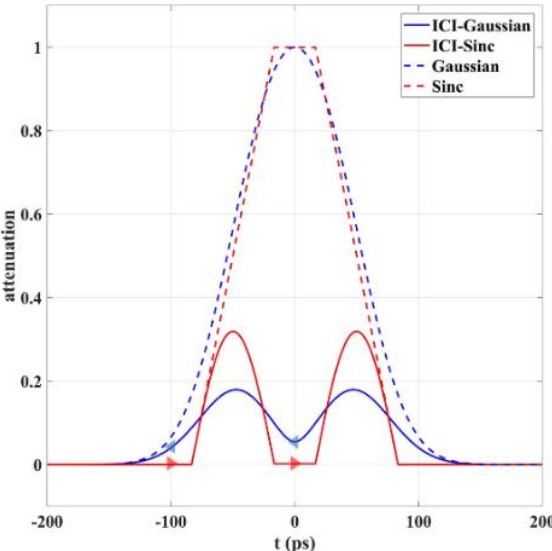

**Figure 4.** Under ideal conditions, the two AO-OFDM systems received the pulse-shaping functions of the corresponding sub-carrier and adjacent sub-carriers at the receiving end of the filter.

The red and blue dashed lines in Figure 4 respectively indicate the pulse-shaping function of a certain sub-carrier of the AO-OFDM system using two filters after passing through the corresponding Sinc-shaped filter with the same center sub-carrier. The red and blue dashed lines respectively indicate the crosstalk of adjacent subcarriers suffered by a single period of a certain subcarrier of the AO-OFDM system using two filters. The red lines represent using a Sinc-shaped filter and the blue lines represent the use of a Gauss-shaped filter.

When *t* = 0, that is, at the optimal sampling point in each symbol period, the filtered AO-OFDM system was sampled, the symbol $C_{ki}$ carried by the target subcarrier was obtained, and the crosstalk $e_{(m-k)}$ from other carriers was also obtained.

$$e_{(m-k)} = \frac{j}{2} \cdot exp\left\{ -\left[2(m-k)^2 f_b{}^2\right]/f_s{}^2 \right\} \left\{ erfi\left[\frac{4(m-k)^2 f_b{}^2 - j\pi f_s}{2\sqrt{2}f_s}\right] - erfi\left[\frac{4(m-k)^2 f_b{}^2 + j\pi f_s}{2\sqrt{2}f_s}\right] \right\} \quad (21)$$

The carrier interference (ICI) generated by the *m*th sub-carrier passing through the *k*th filter, which is the target filter, is shown as the red and blue solid lines. The red and blue triangles in Figure 4 indicate the ICI of the best sampling point in the current symbol period and the best sampling point in the adjacent symbol period. It can be clearly seen in Figure 4 that the AO-OFDM system using a Sinc-shaped filter was not subject to crosstalk between adjacent carriers, while the AO-OFDM system using a Gauss-shaped filter was

subject to crosstalk between carriers. Table 1 shows the influence of CP size and ICI on the current symbol period and adjacent symbol period of the AO-OFDM system based on a Gauss-shaped filter.

**Table 1.** The relationship between CP and adjacent sub-carrier ICI of the AO-OFDM system based on a Gauss-shaped filter.

| CP | 0.09 | 0.17 | 0.23 | 0.29 | 0.33 |
|---|---|---|---|---|---|
| $f_b/f_s$ | 1.1 | 1.2 | 1.3 | 1.4 | 1.5 |
| ICI(0) | 0.144 | 0.071 | 0.014 | 0.028 | 0.054 |
| ICI($\pm T_s$) | 0.049 | 0.047 | 0.044 | 0.042 | 0.040 |

The CP size in the first row of Table 1 corresponds to the ratio of the sub-carrier spacing to the symbol rate after the CP is inserted in the second row. The third row in the table indicates the crosstalk of adjacent sub-carriers to the current sub-carrier at the optimal sampling point $t = 0$. The fourth row represents the crosstalk of adjacent subcarriers to the current subcarrier for the two adjacent sampling points $t = \pm T_s$ of the optimal sampling point. We can find from Table 1 that the crosstalk between carriers was the smallest when the inserted CP size was 0.23. The size of ICI was not directly proportional to the size of CP. However, this situation will become complicated under the influence of chromatic dispersion. According to the simulation results, the larger the CP inserted, the better the anti-dispersion ability of the AO-OFDM based on the Gauss-shaped filter and Sinc-shaped filter.

In Ref. [8], the ICI and Inter-Symbol Interference (ISI) of the AO-OFDM system based on the Gauss-shaped filter were not affected by dispersion, and the frequency domain waveform of the Gauss-shaped filter is discussed in this article. Compared with Ref. [8], it was closer to a Sinc-shaped filter, and we already made a formula derivation, so we will not discuss the relationship between the ICI, ISI, and CP of the system without dispersion. We will mainly discuss why the AO-OFDM system based on the Gauss-shaped filter was more resistant to dispersion.

The spatio-temporal complex envelope (STCE) of a single symbol period of the kth subcarrier of the AO-OFDM signal along the fiber at position $z$ and STCE at time $t$ satisfies NLSE [18]:

$$\frac{\partial g_k(z,t)}{\partial z} - j\frac{\beta''}{2}\frac{\partial^2 g_k(z,t)}{\partial t^2} + \frac{\alpha}{2}g(z,t) = j\gamma|g_k(z,t)|^2 g_k(z,t), t \to t - \beta'z \tag{22}$$

where $\alpha$ is the loss coefficient, $\gamma$ is the nonlinear coefficient, and $\beta' \equiv \partial\beta(\omega)/\partial\omega$ is the first derivative of the transmission constant, which reflects that the difference in group velocity between sub-carriers caused the walk-off between carriers. As the sub-carrier spacing in the AO-OFDM system was small and the deviation was not obvious, the envelope was time-shifted with the group velocity during analysis, and only the second derivative of the transmission constant, that is, the influence of the dispersion on the system, was considered. $\beta'' \equiv \partial^2\beta(\omega)/\partial\omega^2$ is the second derivative of the transmission constant, also known as the second-order dispersion effect, which is one of the main causes of signal envelope distortion and is also the main factor affecting the signal transmission quality of the AO-OFDM system mentioned in Section 1. If only considering the influence of chromatic dispersion on the signal, Equation (22) can be expressed as:

$$\frac{\partial g_k(z,t)}{\partial z} - j\frac{\beta''}{2}\frac{\partial^2 g_k(z,t)}{\partial t^2} = 0, t \to t - \beta'z \tag{23}$$

Transforming Equation (23) into the frequency domain can be expressed as:

$$j\frac{\partial G_k(z,\Omega_k)}{\partial z} = \frac{1}{2}\beta''\Omega^2 G_k(z,\Omega_k) \tag{24}$$

In Equation (24), $\Omega_k = 2\pi(f - f_k)$, then the frequency domain waveform of the signal after the transmission distance $z$ is solved as:

$$G_k(z, \Omega_k) = G_k(0, \Omega_k)exp(-\frac{j}{2}\beta''\Omega_k{}^2 z) \tag{25}$$

$$g_k(z, t) = \int_{-\infty}^{\infty} g_k(0, \Omega_k)exp(-\frac{j}{2}\beta''\Omega_k{}^2 z)exp(j2\pi ft)df \tag{26}$$

The AO-OFDM system pulse-shaping function $g'_{cp}(0,t)$ based on Gauss-shaped filtering is substituted into Equation (26) to obtain the pulse-shaping function $g'_{cp}(z,t)$ after its pulse-shaping symbol is subjected to dispersion.

$$g'_{cp}(0, t) = \sqrt{\frac{\pi}{2}} \cdot exp(-\frac{\pi^2 t^2}{2T_s^2}) \tag{27}$$

$$g'_{cp}(z, t) = \sqrt{\frac{\pi}{2}} \cdot \left(\frac{T_s^2}{T_s^2 + j\beta''\pi^2 z}\right)^{\frac{1}{2}} exp\left(-\frac{\pi^2 t^2}{2(T_s^2 + j\beta''\pi^2 z)}\right) \tag{28}$$

$$g'_{cp}(z, t) = (1 + z^2/L_D{}^2)^{\frac{1}{4}} exp\left[-\frac{\pi^2 t^2}{2T_s^2(1 + z^2/L_D{}^2)^2}\right]exp[j\varphi(z, t)]$$

$$L_D = T_s^2/(|\beta''|\pi^2)$$

$$\varphi(z, t) = \frac{\beta''\pi^2 z}{2T_s^2(1 + z^2/L_D{}^2)^2} \cdot \frac{\pi^2 t^2}{T_s^2} - \frac{1}{2}\arctan(\frac{\beta''\pi^2 z}{T_s^2}) \tag{29}$$

After completing Equation (28), the intensity and phase of the pulse-shaping function can be obtained, where $L_D$ is the dispersion length. We found that the pulse-shaping function produced by the Gauss-shaped filter was still a Gaussian function, even after transmission through the optical fiber, with only broadening and phase changes, and the degree of distortion was much smaller than the pulse-shaping function produced by the Sinc-shaped filter. When the transmission distance was much smaller than the dispersion length, there would be no inter-symbol crosstalk due to dispersion. The pulse-shaping function produced by the Sinc-shaped filter was affected by dispersion. The function is difficult to express with mathematical formulas. We will analyze it from Figures 5 and 6.

Figure 5 shows the pulse-shaping function $g(t)$ of an AO-OFDM signal based on a Sinc-shaped filter affected by dispersion after being transmitted in optical fibers of different lengths. The symbol period was 100 ps, $\beta'' = -20\text{ps}^2/\text{km}$. We can see from the change in the pulse-shaping function after being affected by different accumulative dispersions that chromatic dispersion had a great influence on the pulse-shaping function of the symbol, and these distortions were irregular. At $t = 100$ ps, the amplitude of the optimal sampling point in the adjacent period changed drastically. Even if the accumulative dispersion was 200 ps/nm, the ISI was already close to 0.1. And the size of ISI did not change regularly with the transmission distance. We can see from Figure 5 that, as the influence of the dispersion on the system became greater and greater, the energy in each symbol period leaked from the current period to other periods more and more. Figure 6 is the pulse-shaping function based on the Gauss-shaped filter as the comparison group of Figure 5 under the influence of chromatic dispersion, which transmitted different distances in the optical fiber.

From Equation (29) and Figure 6, we can see that the intensity of the pulse-shaping function generated by the Gauss-shaped filter expanded as the influence of the dispersion increased. Compared with the pulse-shaping function based on the Sinc-shaped filter, the influence of the dispersion at the three sampling points of $t = 0$, $-T_s(-100$ ps$)$, and $T_s(100 \text{ ps})_s$ was small, and the pulse-shaping function did not have a large distortion.

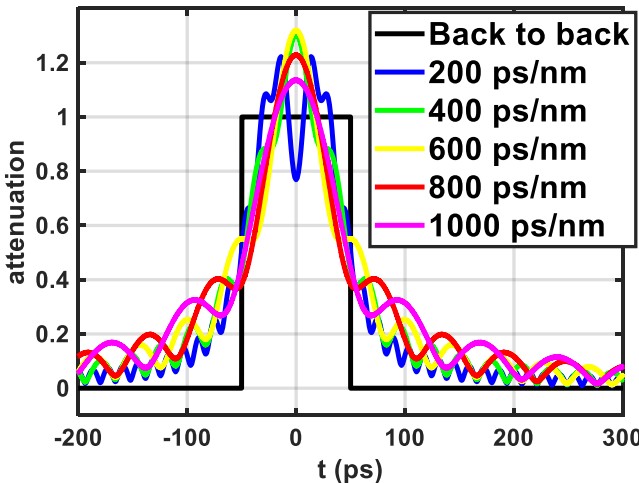

**Figure 5.** Change in $g(t)$ affected by different accumulative dispersions.

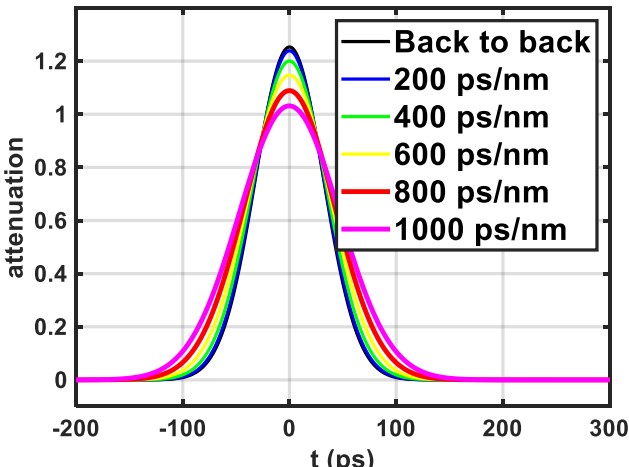

**Figure 6.** Change in $g'_{cp}(t)$ affected by different accumulative dispersions.

Figure 7 shows the pulse-shaping function of a certain sub-carrier through the Sinc-shaped filter at the receiving end after being affected by 800 ps/nm accumulative dispersion and the pulse-shaping function of adjacent carrier crosstalk. Similar to Figure 4, we normalized the received signal. The symbol period was 100 ps, CP = 0.33, and the second-order dispersion coefficient was $\beta'' = -20\text{ps}^2/\text{km}$. The red triangle to the right and the blue triangle to the left in Figure 7 represent the ICIs of the best sampling points of the two AO-OFDMs in the current symbol period and adjacent cycles. We can see that the crosstalk between the carriers of the AO-OFDM system based on the Sinc-shaped filter was much greater than the crosstalk between the carriers of the AO-OFDM system based on the Gauss-shaped filter. In order to more intuitively analyze the relationship between the ICI and the transmitter filter, we converted the CP size into the filter bandwidth and center frequency interval. When the CP was 0.17, 0.23, 0.29, and 0.33, it was converted when the symbol rate was 10 Gbaud, and the bandwidth and center frequency spacing of the Sinc-shaped filter were 12 GHz, 13 GHz, 14 GHz, and 15 GHz, respectively. Although the transmission performance of the system could be improved after the CP was inserted, this also reduced the spectrum efficiency, so we will not discuss the case of inserting a larger CP into the AO-OFDM system.

Figure 8 shows the ICI of the adjacent sub-carrier to the target sub-carrier after the target sub-carrier of the AO-OFDM system with a symbol rate of 10 Gbaud was affected by 800 ps/nm accumulative dispersion. The blue line in Figure 8 represents a system based on a Gauss-shaped filter, and the red line represents a system based on a Sinc-shaped

filter. The dotted line represents the influence of ICI on the optimal sampling point of the current symbol period and the solid line represents the influence of ICI on the optimal sampling point in the adjacent symbol period. On the whole, although the AO-OFDM systems based on Sinc-shaped filters were orthogonal to each other during back-to-back transmission without ICI, their ICI exceeded that of systems using Gauss-shaped filters after being affected by dispersion. Figure 8 shows that, although the size of the inserted CP could affect the ICI of the AO-OFDM system, the ICI of the AO-OFDM system based on the Gauss-shaped filter was always smaller than that of the system based on the Sinc-shaped filter after 40 km of transmission.

In the next part, we will explain the superiority of the Gauss-shaped filter over the Sinc-shaped filter from another aspect.

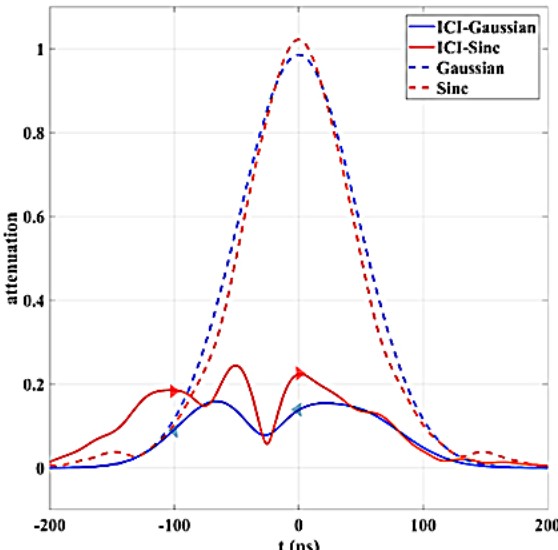

**Figure 7.** After being affected by 800 ps/nm accumulative dispersion, the filter of the two AO-OFDM systems at the receiving end received the pulse-shaping function of the corresponding sub-carrier and the adjacent sub-carrier.

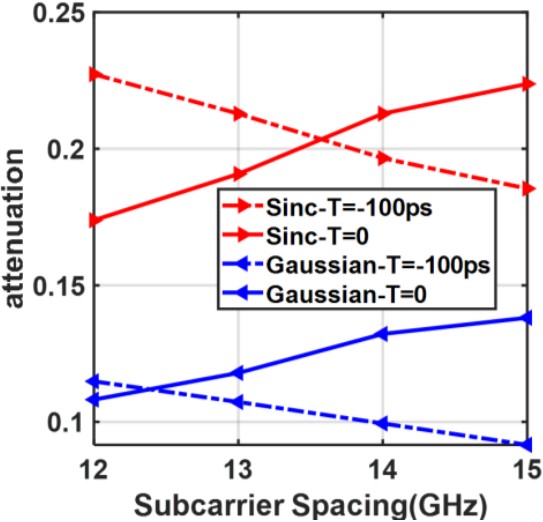

**Figure 8.** The ICI size of adjacent sub-carriers of AO-OFDM signals based on two filters with carrier spacing of 12, 13, 14, and 15 (CP = 0.17, 0.23, 0.29, 0, and 33) after being affected by 800 ps/nm accumulative dispersion.

## 3. Simulation and Results

In this section, we will further demonstrate the superiority of the proposed scheme in terms of anti-dispersion performance through the two different filter AO-OFDM systems' transmission simulation. By analyzing the bit error rate of the AO-OFDM system with 8, 16, and 32 carrier numbers after 10 to 80 km of SMF transmission without dispersion compensation, it is possible to clearly compare the anti-dispersion capabilities of the two AO-OFDM systems. We used the iterative split-step Fourier method to simulate the chromatic dispersion and nonlinear effects of optical signal transmission in the optical fiber. When considering nonlinear effects, we considered self-phase modulation, cross-phase modulation, and four-wave mixing effects. When considering the influence of dispersion on the system, we not only analyzed the second-order dispersion coefficient, but also considered the walk-off effect of the sub-carriers produced by the first-order dispersion coefficient. CPs of different sizes were inserted, and the symbol rate of the $4-$level pulse-amplitude-modulation (PAM-4) signals or 16-quadrature amplitude modulation (16 QAM) signals carried by the subcarriers was 10G baud with a signal length of $2^{17} - 1$. The power of each sub-carrier was $-10$ dBm. Figure 1 shows the simulation of AO-OFDM system setup. An optical frequency comb with 10 GHz spacing was generated by a MLL and then broadened in a 400 m dispersion-flattened highly nonlinear fiber (DF-HNLF) ($\gamma = 10.7/\text{W/km}$, $\beta'' = -0.446 \text{ ps}^2/\text{km}$, and $\beta''' = 0.0057 \text{ ps}^3/\text{km}$ at 1550 nm). When the iterative split-step Fourier method was used to simulate the transmission of optical signals in the optical fiber, each symbol period was sampled 500 times, the step length was 100 m, and each iteration had five individual steps. The fiber was a standard single-mode fiber (SSMF) with attenuation $\alpha = 0.2$ dB/km, nonlinear coefficient $\gamma = 1.3/\text{W/km}$, and $\beta'' = -20 \text{ ps}^2/\text{km}$. The filter used to separate the sub-carriers at the receiving end was a Sinc-shaped filter, and the bandwidth was the same as the symbol rate of each sub-carrier, which was 10 Gbaud. The center frequency of each filter was the same as the filter corresponding to the transmitting end. The greater the CP inserted, the greater the center frequency interval. The signal underwent clock synchronization and signal demodulation after being converted from an optical signal to an electrical signal by the PD. The dispersion chromatic was left uncompensated at the receiving end in order to reflect the anti-dispersion capabilities of the two systems.

In Figure 9, the dashed and solid lines of pink, light blue, blue, green, and red represent BER with $2^{4-8}$ subcarriers, respectively, for the all-optical OFDM systems based on Sinc-shaped filters and Gauss-shaped filters, respectively, at different accumulated dispersion, and it can be seen that, when the accumulated dispersion was 1000 ps/nm, the BER of an all-optical OFDM system based on a sinc-shaped filter with $2^{4-8}$ subcarriers was higher than $10^{-2}$. When the number of subcarriers was 16, the BER was $4.214 \times 10^{-2}$; when the number of subcarriers was 256, the BER was $6.870 \times 10^{-2}$, and the BER of Gauss-shaped filters-based systems was less than $10^{-2}$. BER $= 4.425 \times 10^{-3}$ when the number of subcarriers was 16, and $7.858 \times 10^{-3}$ when the number of subcarriers was 256. That is to say, when the cumulative CD was 1000 ps/nm, the BER of the all-optical OFDM system based on the Gauss-shaped filter was reduced by more than eight times compared with the system based on the Sinc filter.

In Figure 10, the dashed and solid lines in pink, light blue, blue, and green, respectively, represent the BER of the all-optical OFDM systems with $2^{4-7}$ subcarriers based on the Sinc-shaped filter and the Gauss-shaped filter at different accumulated dispersions, and it can be seen from the overall change trend of the eight curves that, although the modulation format of each subcarrier was different from the PAM4 in Figure 10, the overall change trend was almost the same. When the accumulated dispersion was 1000 ps/nm, the BER of an all-optical OFDM system based on a Sinc-shaped filter with $2^{4-7}$ subcarriers was higher than $5 \times 10^{-2}$, where the number of subcarriers was 16, BER $= 7.081 \times 10^{-2}$. The BER of the Gauss-shaped filter-based system was less than $5 \times 10^{-2}$, BER $= 3.113 \times 10^{-2}$ when the number of subcarriers was 16, and BER $= 3.956 \times 10^{-2}$ when the number of subcarriers was 128.

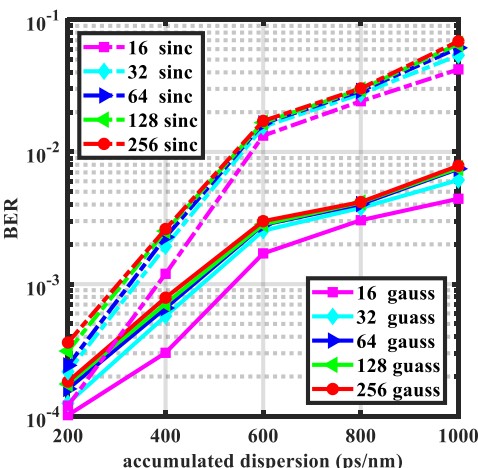

**Figure 9.** BER of two all-optical OFDM systems based on different filters with PAM−4 subcarrier modulation format after being affected by accumulated dispersion of different sizes when CP is not inserted.

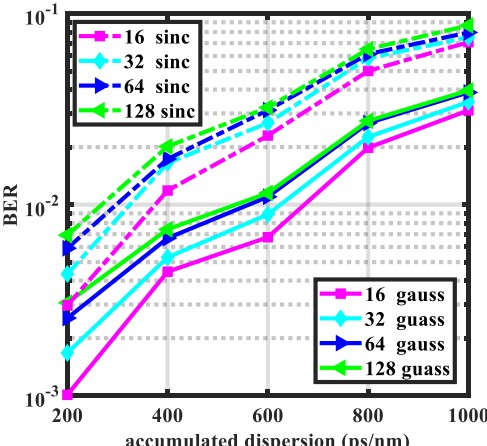

**Figure 10.** BER of two all-optical OFDM systems based on different filters with 16 QAM subcarrier modulation format after being affected by accumulated dispersion of different sizes when CP is not inserted.

As can be seen from Figures 9 and 10, when the Gauss-shaped filter replaced the Sinc-shaped filter, the BER of the system was greatly reduced under the influence of different accumulated dispersions. However, the reduction was not large enough, and it was necessary to carry out frequency domain sparsening treatment on the subcarrier, which could also be said to insert CP into the all-optical OFDM system to reduce the crosstalk caused by the insufficient orthogonality of the subcarrier generated by the Gauss-shaped filter, and further highlighted its anti-dispersion ability. Figure 11 shows two all-optical OFDM systems with a subcarrier power of −13 dBm, modulation format of 16 QAM, and 32 orthogonal subcarriers, and the BER varied with transmission distance after inserting CP = 0 and CP with a size of 0.33.

In the modulation format 16 QAM of the system in Figure 12, for the all-optical OFDM system with a Sinc filter before inserting CP, BER = $8.545 \times 10^{-2}$ after 60 km transmission; after inserting CP of size 0.33, BER = $1.925 \times 10^{-2}$; and after replacing the Sinc filter with a Gaussian filter, BER = $1.596 \times 10^{-3}$. The two constellations in Figure 12 more clearly reflect the difference between the use of a Sinc filter and a Gaussian filter in the AO-OFDM system, and the change in the AO-OFDM signal transmission quality before and after CP insertion.

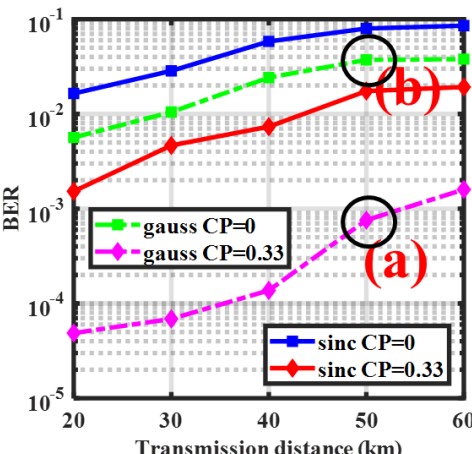

**Figure 11.** Two all−optical OFDM systems based on different filters with 16 QAM subcarrier modulation format with the change in BER with transmission distance after inserting CP = 0 and CP with a size of 0.33. Two points (**a**,**b**) in the figure correspond to the two constellation diagrams in Figure 12.

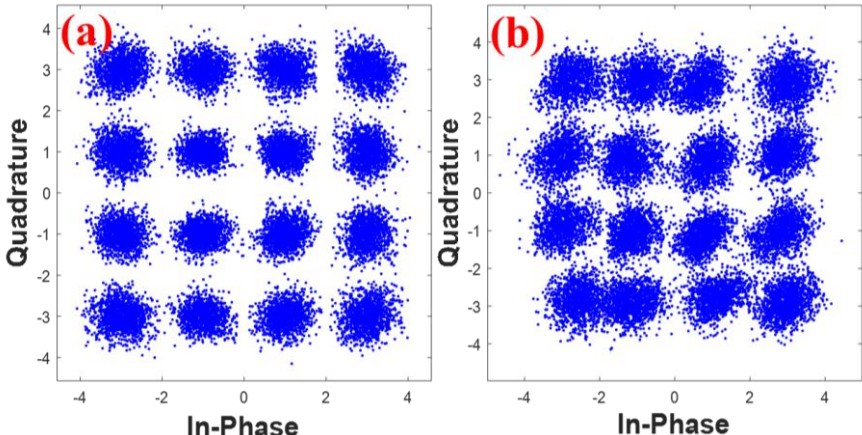

**Figure 12.** In Figure 11, the two points (**a**,**b**) correspond to these constellation diagrams.

## 4. Conclusions

We proposed a new method to improve the anti-dispersion ability of all-optical orthogonal frequency-division multiplexing (AO-OFDM) systems. By replacing the Sinc-shaped filter with a Gauss-shaped filter to generate a subcarrier, combined with the insertion of CP, the effect of dispersion on the system could be significantly reduced. Compared with the method mentioned in Ref. [7], the proposed scheme could minimize the overhead of the AO-OFDM signal time domain without reducing the duration of the signal in each time period, and at the same time, made the signal achieve better anti-dispersion ability. The simulation results show that, in an all-optical OFDM system with 32 subcarriers and a modulation format of QPSK, BER = $8.545 \times 10^{-2}$ after 60 km transmission without a Sinc-shaped filter and no CP being inserted, and the Sinc-shaped filter was replaced with a Gauss-shaped filter and an optical CP with a size of 0.33 was inserted into the optical CP; that is, after increasing the subcarrier spacing by 1.5 times, BER = $1.596 \times 10^{-3}$, and BER decreased by more than 50 times.

**Author Contributions:** Conceptualization, K.L. and C.Y.; methodology, C.Y.; software, L.F.; validation, X.S., K.L. and C.Y.; formal analysis, X.W.; investigation, X.W.; resources, C.Y.; data curation, A.Z.; writing—original draft preparation, H.L.; writing—review and editing, C.Y.; visualization, K.L.; supervision, Y.L.; project administration, Y.L. All authors have read and agreed to the published version of the manuscript.

**Funding:** This research received no external funding.

**Institutional Review Board Statement:** Not applicable.

**Informed Consent Statement:** Not applicable.

**Data Availability Statement:** The data underlying the results presented in this paper are not publicly available at this time, but may be obtained from the authors upon reasonable request.

**Conflicts of Interest:** The authors declare no conflict of interest.

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
