# Peer review of "Enhancing the Anti-Dispersion Capability of the AO-OFDM System via a Well-Designed Optical Filter at the Transmitter"

_photonics, doi:10.3390/photonics10091053_

Round 1

Reviewer 1 Report

Im very glad to review the paper in greater depth because the subject is interesting. And the submission is worth of publication. Following are some minor comments:

There are several abbreviation errors in the article, such as ”mode-locked lasers (MML)”, that need to be corrected.

Can the method described in the article to improve dispersion resistance be applied to other OFDM systems?

The “7% HD-FEC” already appears in Figure 9 in the article, but its abbreviations does not appear in the description of Figure 9, and the author needs to correct it.

Please explain why the sample rate of the simulation is as high as 500 samples per symbol

none

Author Response

Response to Reviewer 1 Comments

Point 1: There are several abbreviation errors in the article, such as ”mode-locked lasers (MML)”, that need to be corrected.

Response 1: I would like to sincerely apologize for our inadequate description. We made revisions to the manuscript

Point 2: Can the method described in the article to improve dispersion resistance be applied to other OFDM systems?

Response 2: The proposed solution may only work for all-optical OFDM systems, because the waveform function of each subcarrier needs to be changed, which needs to be implemented by an optical filter when generating the subcarrier. In coherent optical OFDM systems, the OFDM signal is generated in the electrical domain, and it is difficult to change the waveform of each subcarrier through the optical filter.

Point 3: .The “7% HD-FEC” already appears in Figure 9 in the article, but its abbreviations does not appear in the description of Figure 9, and the author needs to correct it.

Response 3: I would like to sincerely apologize for our inadequate description. We made revisions to the manuscript

Point 4: Please explain why the sample rate of the simulation is as high as 500 samples per symbol

Response 4: Thank you very much for your valuable suggestions. Analysis of the entire AO-OFDM signal after being affected by dispersion needs to consider the effect of filtering at the receiving end, it is the entire AO-OFDM signal filtered, so the larger sampling rate ensures that the information carried by each subcarrier will not be lost during the filtering process, and the higher sampling rate ensures the accuracy of the simulation.

Reviewer 2 Report

The author presented good work in enhancing the anti-dispersion capability of an all-optical OFDM system based on two types of optical filters. The results are satisfactory. However, there are some concerns that should be addressed:

1)      The manuscript should be rewritten with more vivid work.

2)      The novelty of this paper over prior arts is not clear.

3)      The figures should be redrawn in a good way (resolution, legend, the size of figures is not proportional to text size, etc).

4)      The equations should be rewritten in a more advanced way.

5)      More references should be added to the list.

6)      What are the definition and characteristics of CD?

7)      There are some grammatical issues in the text and some typos.

8)      Can the author display the constellation diagrams as well as EVM values as a function of transmission distance in order to know how much the constellation diagram differs from WSS transmitter to WSS receiver?

9)      Only the BER curve is shown. Can the authors comment on the EVM curve´s performance over OSNR?

10)  Is this architecture only working for QPSK? How about the performance for 16-QAM?

11)  Why is the FEC threshold only 7% for the OFDM-QPSK transmission?

There are some grammatical issues in the text and some typos. Try to modify the English quality of the paper.

Author Response

Response to Reviewer 2 Comments

Point 1: The manuscript should be rewritten with more vivid work.

Response 1: Thank you for your valuable advice, we have made a lot of revisions to the manuscript

Point 2: The novelty of this paper over prior arts is not clear.

Response 2: I would like to sincerely apologize for our inadequate description. Our work is to improve the anti-dispersion ability of the AO-OFDM system by modifying the filter function, which is completely different from other work.

Point 3: The figures should be redrawn in a good way (resolution, legend, the size of figures is not proportional to text size, etc).

Response 3: I would like to sincerely apologize for our inadequate description. We submitted new figures in the newly submitted manuscript..

Point 4: The equations should be rewritten in a more advanced way.

Response 4: Thanks for the suggestion, we modified the formula in the article

Point 5: More references should be added to the list.

Response 5: Thanks for the suggestion, We adjusted the reference list.

Point 6: What are the definition and characteristics of CD?

Response 5: I would like to sincerely apologize for our inadequate description. CD stands for Chromatic dispersion. In the previously submitted manuscript, there is an error in the formulation of the simulation section, "The CD is left uncompensated at the receiving end in order to reflect the anti-dispersion capabilities of the two systems." The "CD" in this sentence is our typo and should be "dispersion compensation", which we have corrected in the newly submitted manuscript.

Point 7: There are some grammatical issues in the text and some typos.

Response 6: I would like to sincerely apologize for our inadequate description.We corrected these spelling and grammatical errors in the newly submitted manuscript

Point 8: Can the author display the constellation diagrams as well as EVM values as a function of transmission distance in order to know how much the constellation diagram differs from WSS transmitter to WSS receiver?

Response 8: Thank you very much for your suggestion, we have added constellation diagrams to the new manuscript submission to more clearly reflect the advantages of our proposed anti-dispersion scheme.

Point 9: Only the BER curve is shown. Can the authors comment on the EVM curve´s performance over OSNR?

Response 9: Thank you for your suggestion, our current main work focuses on the influence of filter waveform on the anti-dispersion performance of AO-OFDM system, and the change of system transmission quality can best be reflected by BER. We will analyze and study your suggestions in the next work, thank you again for your suggestions.

Point 10: Only the BER curve is shown. Can the authors comment on the EVM curve´s performance over OSNR?Is this architecture only working for QPSK? How about the performance for 16-QAM?

Response 10: Thanks for your suggestion, we have added an analysis of the performance when the modulation format of each subcarrier is 16-QAM

Point 11: Why is the FEC threshold only 7% for the OFDM-QPSK transmission?

Response 11: Unfortunately, we can't answer your question, but we use the 7% HD-FEC threshold in the references we cited, so we use it as well